# Drop Weight Testing of Samples Made of Different Building Materials Designed for the Protection of Classified Information

**DOI:** 10.3390/ma16031219

**Published:** 2023-01-31

**Authors:** Jakub Durica, Andrej Velas, Martin Boros, Radoslav Sovjak, Petr Konrad, Premysl Kheml

**Affiliations:** 1Department of Security Management, Faculty of Security Engineering, University of Zilina, 010 26 Zilina, Slovakia; 2Experimental Centre, Faculty of Civil Engineering, Czech Technical University in Prague, 150 00 Prague, Czech Republic

**Keywords:** classified information, resistance, wall, drop weight testing, impact

## Abstract

Classified information is information of vital importance to the State, which must be protected against disclosure, misuse, damage, unauthorized reproduction, destruction, loss or theft in the interest of the State. At present, there are four levels of classification. For each classification level, precise requirements are laid down for the material of the walls, partitions and ceilings of the rooms in which classified information is stored. Several types of materials are defined for each classification level. The objective of this study was to test and determine whether the different types of materials proposed for the Confidential level meet the same level of resistance. A drop weight test via pendulum was used to determine the resistance. A 50 kg weight was used to break through a 60 × 100 cm sample. The impact of the weight was on the exact center of the sample. The result of the tests was that to break through samples of different materials, large differences in the drop height of the weight were required. The most resistant was the specimen made of reinforced concrete, which required 3 impacts from a height of 80 cm to break through. On the contrary, the least resistant were the specimens made of masonry of autoclaved aerated concrete, where after 2 falls from a height of 5 cm, the sample broke into 2 parts.

## 1. Introduction

Classified information is information of vital importance to the state which, in the interests of the state, must be protected against disclosure, misuse, damage, unauthorized reproduction, destruction, loss or theft. In the Slovak Republic, classified information is defined in Act 215/2004 on the Protection of Classified Information. According to Act 215/2004, classified information is information or matter determined by the originator of the classified information which, in view of the interest of the Slovak Republic, must be protected against disclosure, misuse, damage, unauthorized reproduction, destruction, loss or theft and which may only arise in the areas established by the Government of the Slovak Republic by its regulation [1]. In the United States, classified information is defined as information that, for reasons of national security, is specifically designated by a United States Government agency for restricted or limited dissemination or distribution [2]. In Poland, classified information is defined as information disclosure that would or could cause damage to Poland or would be disadvantageous for Poland [3]. In the Czech Republic, classified information is information in any form recorded on any medium marked in accordance with Act 412/2005, the disclosure or misuse of which may cause harm to the interest of the Czech Republic or may be disadvantageous to that interest, and which is included in the list of classified information [4]. European Union Classified Information (EUCI) means any information or material classified at the EU security level, the unauthorized disclosure of which could cause varying degrees of damage to the interests of the European Union or of one or more Member States [5].

The European Union and the Member States divide classified information into four categories [1,5]:Top secret—information and material the unauthorized publication of which could cause extremely serious damage to the essential interests of the European Union or of one or more of its Member StatesSecret—information and material the unauthorized disclosure of which could seriously damage the essential interests of the European Union or of one or more Member StatesConfidential—information and material the unauthorized disclosure of which could damage the essential interests of the European Union or of one or more Member StatesRestricted—information and material the unauthorized disclosure of which could be disadvantageous to the interests of the European Union or of one or more Member States

Classified information is protected by a series of security procedures based on its designation [1]. One of these procedures is to secure rooms or whole buildings with appropriate security barriers [6,7]. At present, there is no established name for security barriers, so it is possible to come across names such as physical security barriers, physical barriers or just barriers [8,9,10]. In this article, the term security barriers will be used. Security barriers consist of a set of mechanical and technical means, devices and components which, by their design, make it impossible to break through them easily. In terms of evaluating a physical security system, it is possible to benefit from knowing the resilience and time to break through each barrier [7,11]. Security barriers that, on the one hand, reduce the possibility of burglary, but on the other hand, can reduce the aesthetic value of the building, for example, in the target hardening concept, where they contribute to the deterrent effect of the entire security system. In simple terms, it can be said that security barriers are obstacles that are placed between the intruder and the protected interest [12,13].

### Security Barriers—Walls

The perimeter walls are the most important layer of protection of the building because by overcoming them, the attacker can get inside the building. Quite often, mostly in urban agglomerations, it is shell protection that forms the first barrier between the attacker and the protected interest [11,14,15].

The building elements of the buildings are security barriers protecting the envelope. Building elements include the building envelope walls, ceiling and roof. Depending on the level of security or break-through resistance, buildings are divided into [7,15,16]:Lightweight buildings—are those buildings where the penetration resistance is low or the time of penetration resistance is short; they are used for enclosing the space. These include plasterboard or plasterboard partitions and fillings, walls and partitions made of autoclaved aerated concrete blocks, partitions and walls made of chipboard, wool and steel sheets, concrete walls without reinforcement up to 50 mm, etc.Solid building structures—are characterized by high break-through resistance. Materials such as concrete, stone, reinforced concrete blocks, brick, etc., are used for solid structures.

So far, several tests have been carried out to determine the break-through resistance of the security walls. Of note is the test where panel walls 3rd Resistance Class, and 4th Resistance Class were tested, i.e., for Secret and Top Secret, Figure 1. The test methodology was based on EN 1627 and EN 1630. The primary purpose of the test was to create a break-through opening to determine the delay time [17].

A similar test was carried out at the Certest testing laboratory commissioned by the National Security Authority of Slovakia. The methodology of the test was carried out on the basis of the Test Code—Burglary Resistance of Building Materials CTSP 01/2005 MET 11/2005. Several types of materials were tested, such as masonry made of Ceramic blocks, gypsum plasterboard partition, and masonry made of aerated concrete, Figure 2. However, they do not specify which security class it is. The static test and resistance to manual burglary attempts were also carried out [18].

This kind of testing is very subjective due to the fact that the test is performed by a human, and the time of breaking depends on his experience and skills.

Weight-breaking tests, i.e., drop weight testing, have been performed on a sample of material. As an example, Gunasekaran et al. [19] investigated the impact toughness of two concrete mixtures through a drop weight test. The tests were carried out using the procedure proposed by the ACI 544 committee. Results for both concrete mixes are approximately equal. Another test was conducted by Jabir et al. [20] to analyze different types of ultra high-performance concrete. The test results showed that the mixes with 15 mm micro-steel fibers absorbed a higher number of impact blows until cracking occurred compared to the other mixes. The mix with 2.5 volume 15 mm micro-steel fiber showed the highest impact resistance, with the percentage increase over the other mixes ranging from 25% to 140%. Experimental testing of self-compacting concrete enhanced with steel fibers through a drop weight test was also carried out by Abid et al. [21]. The testing consisted of the impact of a 5.47 kg free-falling body from a height of 100 mm. The test results showed that the impact resistance and ductility were significantly improved due to the incorporation of micro-steel fibers. The percentage improvements were significantly greater in the failure phase than in the cracking phase. For the 30 MPa mixtures, the maximum percentage improvements were 543% in the cracking phase and 836% in the failure phase. It is also worth mentioning the test by Murali et al. [22], where the ACI 544 standard was tested but with minor changes. The first change was to replace the steel ball with a steel bar to use a line impact instead of a single-point impact. The second and third introduced linear and cross notches on the top surface of the specimen and applied loads through the steel plate of the cross knife or linear load type. These modifications spread the impact load over a larger area and reduced the scatter of the results. The fourth and fifth were a bed of sand and coarse aggregate as an alternative to a solid foundation slab. One hundred and eight cylindrical specimens were prepared and tested in 12 groups to evaluate the proposed modification methods. The weights were 4.5, 6.0 and 7.5 kg, and the drop heights were 450, 575 and 700 mm. The test results are that the specimens with linear notches and sand bedrock significantly reduced the coefficient of variability of the test results, indicating some changes.

Based on the above results in this area, the conclusion was formulated that although the influence of the human conducting the tests has been removed. However, the dimensions of the specimens were very small to demonstrate whether walls made of such materials are resistant to overcoming. Based on the research, it was chosen as the objective of this study to test the durability of wall samples that are used to protect classified information. In the durability tests, the influence of the penetrator will be eliminated through the impact pendulum. The reason for these tests is that no similar type of tests have been conducted before if the impact of the overtaker would be eliminated, but also that currently, for one security class, there are several types of materials that can be used to construct this wall.

## 2. Materials and Methods

As has already been mentioned, there is currently no single common procedure or regulation governing the protection of classified information. In the Slovak Republic, Decree No. 336/2004 of the National Security Authority applies, together with an addendum that sets out the construction materials for walls, partitions and ceilings [23]. It was therefore decided to test the resistance of the materials described in this appendix. Specifically, the 2nd Resistant Class of security, which is equivalent to Confidential, was tested. The material specification for Confidential can be found in Table 1.

It was decided to test only samples and not entire walls or wall systems. A total of 3 identical samples were made from each material. The sample had dimensions of 100 cm in length and 60 cm in height, and the thickness is described in Table 1, different for each construction material. Individual samples can be seen in Figure 3.

Impact pendulum has been used to test for resilience, Figure 4. A 50 kg weight was used to load the specimen. PCB Piezotronics 350B04 accelerometers were placed on the upper and lower parts of the impactor. Accelerations were recorded at a frequency of 500 kHz. The measured data were filtered using a CFC 600 filter, averaged between the two accelerometers, and clipped from the moment of impact to the steady state.

The dimensions of the 60 × 100 cm of each sample were such that the weights dropped exactly in the center.

The testing methodology was inspired by EN 1629 and EN 1630. The aim was to create a hole that matched the shape of the weight, i.e., the weight penetrated the sample. When 3 test samples of each material were made, 3 different tests were performed:gradual increase in drop height (5, 10, 15…cm),maximum drop height,constant drop height.

Constant drop height was called on the basis of the observed results of sample damage in the gradual increase in drop height test. The test was terminated when the above hole was created. The test scheme is shown in Figure 5.

## 3. Results

### 3.1. Masonry of Autoclaved Aerated Concrete

The first test was the masonry of autoclaved aerated concrete, where the maximum fall height from which the masonry breaks were measured. The first impact height was 5 cm. When impacted from this height, there was no visible damage to the specimen. When the weight was dropped from 10 c, the specimen broke into two parts. What was remarkable was that it was the aerated concrete block that was broken and not the point of bonding mortar. The second test consisted of lowering a weight from a height of 30 cm. This test achieved the same results as in the previous test, namely punching and breaking the specimen into two parts. The third and final test consisted of lowering the weight from a constant height until a hole was formed. The drop compliance height was chosen to be 5 cm. Table 2 shows a comparison of the forces for individual drops recorded by the accelerometers.

The result of all the tests was that the specimen broke in two. Surprisingly, it broke at the brick and not at the point of the thin-layer bonding mortar, Figure 6. The reason for the breakage in the brick and not at the point of the thin-layer bonding mortar is that the strength of the thin-layer bonding mortar is greater than the masonry strength of autoclaved aerated concrete.

When testing the resistance of masonry of autoclaved aerated concrete, it has been shown that less than 10 kN is required to break through such a wall, which does not pose much of a challenge to the attacker, and it is assumed that they will break through such a wall.

### 3.2. Masonry Made of Ceramic Blocks

As with the masonry of autoclaved aerated concrete, the first test consisted of a constant increase in fall height. The first fall was carried out from a height of 5 cm, gradually increasing up to a fall height of 30 cm. For the first two falls, i.e., from a height of 5 cm and 10 cm, there was no visible damage to the specimen. The first visible cracks on the plaster were observed from a height of 15 cm. The drop from a height of 30 cm achieved a complete failure by dislodging one ceramic block, which fell out, Figure 7. The second test consisted of dropping the impactor from a height of 50 cm. The result of this test was that the ceramic blocks’ mortar did not hold, and the ceramic blocks flew off. The third and final test consisted of lowering a weight from a constant height of 40 cm. A total of three impacts of the impactor on the masonry were required; however, a hole in the shape of the impactor was not created but, as in the previous tests, the masonry fell out, and the hole created was larger than the dimensions of the impactor. The data from the test can be found in Table 3. The force decreases because the wall has cracked, and therefore its stiffness has decreased.

When testing masonry made of ceramic blocks, it was found that the biggest weakness of such walls is the masonry mortar, as it happened that the masonry mortar loosened and the masonry flew out of its position, Figure 7.

### 3.3. Partition Made of Plasterboard Reinforced with 1 mm Thick Sheet Metal

The resistance testing of the partition made of plasterboard reinforced with 1 mm thick sheet metal was the third in the sequence. The sheet metal is located on the impact side. Based on estimates, it was concluded that the lowest force, i.e., the lowest drop height of the weight, would be required to break through it. The first test was, as with all of them, a gradual increase in drop height. Already at a fall height of 5 cm, minor damage to the plasterboard could be observed. The same was true for all experiments. In the tests, has been found that the plasterboard on both sides broke at a fall height of 15 cm and broke completely at a fall height of 20 cm, but the sheet metal was still not punctured, only dented. Subsequently, the impact did not break the sheet metal, but the frame was damaged and the screws loosened, and the weight went through the entire frame when falling from 30 cm. The second test consisted of dropping the impactor from a height of 50 cm. When dropped from this height, the weight broke through both plasterboards but did not break the sheet metal, but as in the previous case, the bolts did not hold, and the weight went all the way through the test specimen on the other side. For the third test, a fall height of 30 cm was chosen. A total of three falls from this height were required. As in the previous cases, the break-through hole in the shape of the weights failed to form. However, the specimen was destroyed, and the weights passed through the test specimen on the third attempt. The data from the tests is given in Table 4.

When testing 3 partitions made of plasterboard reinforced with 1 mm thick sheet metal, it was concluded that the plasterboards themselves could not withstand a strong impact. However, the sheet metal placed on the attack side resisted all impacts and could not be thwarted; it is questionable how a 1 mm thick sheet metal would withstand an attack with a sharp spike. The results of the tests are shown in Figure 8. The conclusion of the tests is that the weak points of such a wall were the plasterboard itself and the supporting construction when it did not hold up in the tests.

### 3.4. Reinforced Concrete

Last was the testing of reinforced concrete specimens. This was already the case in the first test when gradually increasing the fall height to 55 cm when the concrete had already cracked vertically into two halves, which were held only by the steel reinforcement. However, a hole has been created at the point of impact of the weight, which corresponds to the dimensions of the weight. The recorded force dropped by 15 cm to 35 cm and then rose again. First, the concrete cracked, and the stiffness of the concrete decreased and then the impactor hit the broken concrete, which did not crack further because it was held by the steel. A height of 80 cm was chosen as the maximum drop height due to the limitations of the impact pendulum. A total of three drops were required. Already in the first fall, damage to the sample was visible. In the second fall, a partial break-through hole was already formed, and in the third fall, a part of the sample was knocked out, i.e., a hole of larger dimensions than defined before the start of the testing was managed to form, Figure 9. For the third test, a constant drop height of 55 cm was chosen. A total of six drops were required to create a hole with the dimensions corresponding to the dimensions of the weight, but vertical cracking was observed already at the third drop. As with the previous tests, it should be taken into account that the resulting forces decreased during the experiments as the concrete cracked, and so its stiffness decreased. Data from the reinforced concrete tests are presented in Table 5.

Reinforced concrete samples were the most resistant. During the tests, only a hole in the sample was created. However, the steel reinforcement played a major role in holding the concrete and thus the whole specimen together even after several impacts, when a 1 cm wide crack ran through the concrete along its entire vertical length, but even when the impactor knocked out a large volume of material.

## 4. Discussion and Conclusions

The results of the tests showed that although they are the same security class, i.e., walls, partitions and ceilings designed for the Confident classified level, there are large differences between the different materials. It would therefore be necessary to standardize this methodology internationally, either within the European Union or NATO.

The aim of our study was to test the walls, partitions and ceilings of buildings containing classified information. The survey found that each country has its own procedure for protecting classified information. The materials and detailed specifications used in the Slovak Republic were tested. The test methodology was inspired by EN 1629 and EN 1630, but also ACI 544. The conclusions and summary of the results of the drop weight test for materials intended for the protection of classified information of confidential level are as follows:In terms of puncture resistance, aerated concrete, which cracked in the brick and broke in two pieces in all three tests, was the least resistant. Reinforced concrete was the most durable, but the concrete itself cracked and formed a hole even though it was still held together by reinforcement. The weakness of the masonry of ceramic blocks was the masonry mortar, which complete failure and the ceramic block, which fell out. It was characteristic of the plasterboard partition specimens reinforced with 1 mm thick sheet metal on the attack side that in neither test method did the impactor penetrate the sheet metal, but the plasterboards were broken on both sides, and the bolts reinforcing the construction did not hold.A future outlook for a possible methodology for testing security walls. The shape and weight of the impactor need to be considered, as the impactor did not penetrate the 1 mm thick sheet metal with the rounded tip impactor but assume that penetration would not be a problem with the sharp tip impactor.The measured force does not constantly increase as the drop height increases. However, accelerometer measurements can give us information about the maximum force and correspond to the loss of stiffness of the samples.The test results were surprising to us as they are the same security class. They should therefore have approximately the same resistance. However, this does not account for the fact that the aerated concrete specimen broke in two parts after two impacts from a height of 5 cm and that it took three impacts from a height of 80 cm to break the reinforced concrete.

## Figures and Tables

**Figure 1 materials-16-01219-f001:**
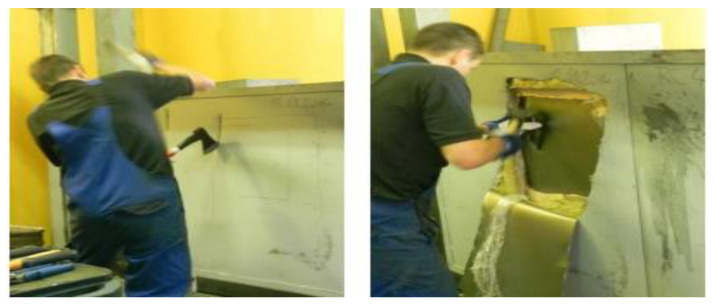
4th Resistance Class panel walls break-through testing [17].

**Figure 2 materials-16-01219-f002:**
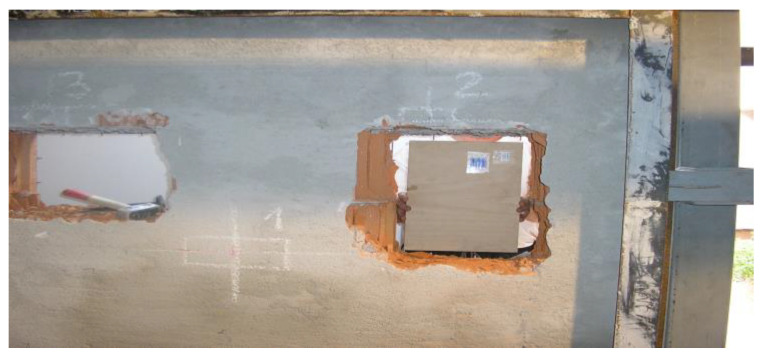
Breaking test of masonry made of ceramic blocks [18].

**Figure 3 materials-16-01219-f003:**
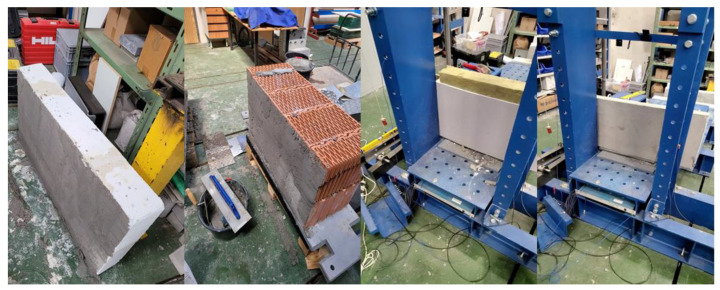
Testing samples, from the left—aerated concrete, ceramic brick, plasterboard, reinforced concrete.

**Figure 4 materials-16-01219-f004:**
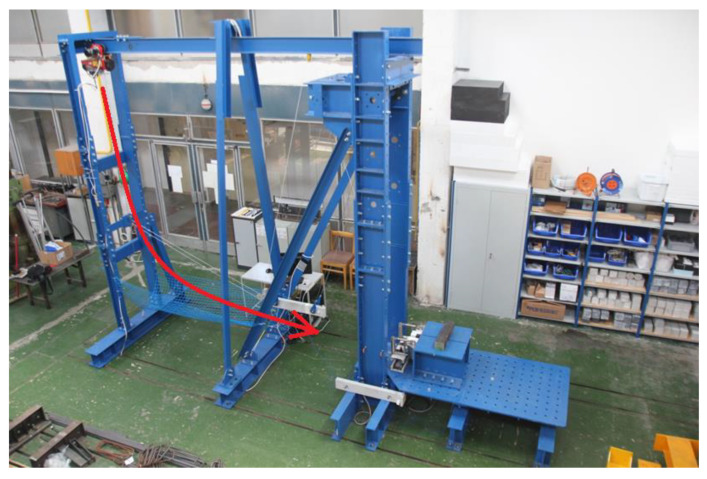
Pendulum.

**Figure 5 materials-16-01219-f005:**
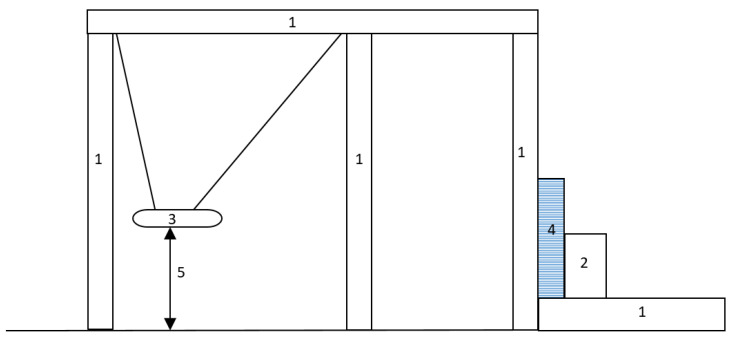
Pendulum diagram: 1—pendulum, 2—stabilizing weight, 3—drop weight, 4—specimen, 5—height.

**Figure 6 materials-16-01219-f006:**
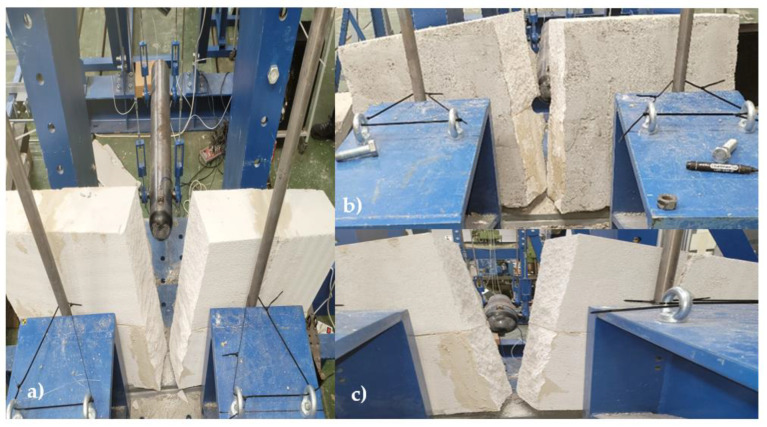
Broken blocks into two parts—(**a**) gradual increase method, (**b**) maximum height method, (**c**) constant height method.

**Figure 7 materials-16-01219-f007:**
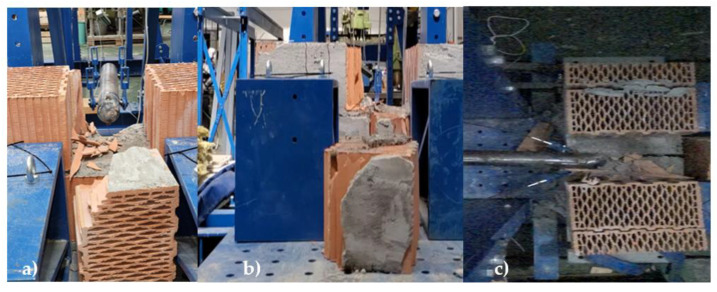
Failure of masonry mortar—(**a**) gradual increase method, (**b**) maximum height method, *(***c**) constant height method.

**Figure 8 materials-16-01219-f008:**
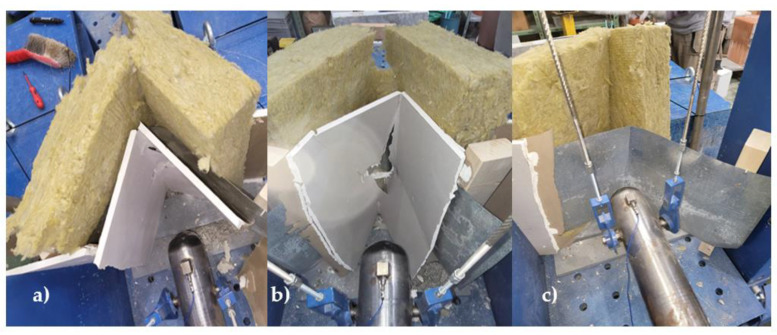
Plasterboard reinforced with 1 mm thick sheet metal testing results—(**a**) gradual increase method, (**b**) maximum height method, (**c**) constant height method.

**Figure 9 materials-16-01219-f009:**
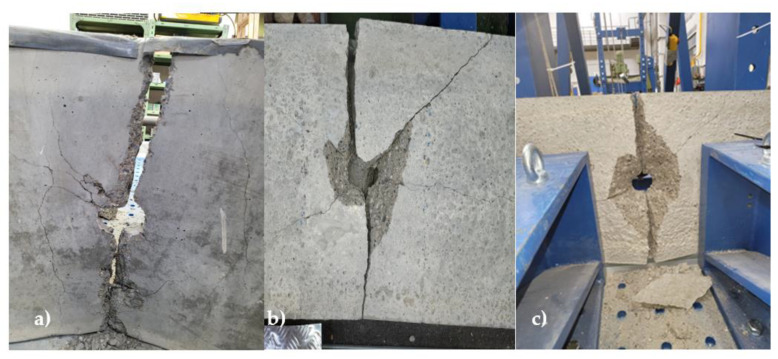
Results of reinforced concrete—(**a**) gradual increase method, (**b**) maximum height method, (**c**) constant height method.

**Table 1 materials-16-01219-t001:** Specification of walls, partitions and ceilings for confidential level [23].

	Confidential			
Construction Material	Thickness [mm]	Building Material	Reinforcement	Note
Masonry of autoclaved aerated concrete	150	Autoclaved aerated concrete blocks	-	Thin-layer bonding mortar
Masonry made of ceramic blocks	400	Ceramic blocks 380 × 250 × 238 mm, 10 MPa	-	Masonry mortar up to 5 MPa, plaster up to 15 mm
Partition made of plasterboard	100	Plasterboard of 12.5 mm thickness with construction	1 mm thick steel plate	Self-tapping screws every 150–200 mm
Reinforced Concrete	75	Concrete C16/20	1 row max 215 mm apart	V10 425 Ø 6 mm in two rows

**Table 2 materials-16-01219-t002:** Data from the masonry of autoclaved aerated concrete testing.

Drop Height (cm)	Force (kN)
gradual increase method	
5	7.67
10	11.55
maximum height method	
30	15.39
constant height method	
5	6.94
5	9.39

**Table 3 materials-16-01219-t003:** Data from the masonry of ceramic blocks testing.

Drop Height (cm)	Force (kN)
gradual increase method	
5	8.05
10	11.66
15	8.62
20	6.09
25	5.15
30	4.80
maximum height method	
50	8.19
Constant height method	
40	21.09
40	12.10
40	4.86

**Table 4 materials-16-01219-t004:** Data from plasterboard reinforced with 1 mm thick sheet metal testing.

Drop Height (cm)	Force (kN)
gradual increase method	
5	0.95
10	0.94
15	1.30
20	1.40
25	1.22
30	1.62
maximum height method	
50	1.87
constant height method	
40	1.36
40	1.25
40	1.94

**Table 5 materials-16-01219-t005:** Data from reinforced concrete testing.

Drop Height (cm)	Force (kN)
gradual increase method	
5	3.87
10	43.72
15	40.24
20	32.47
25	31.63
30	34.13
35	17.86
40	28.05
45	26.67
50	37.02
55	30.25
maximum height method	
80	104.80
80	53.06
80	35.38
constant height method	
55	90.70
55	45.69
55	41.05
55	33.12
55	34.26
55	35.00

## Data Availability

Data sharing is not applicable to this article.

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
