# Peer review of "Drop Weight Testing of Samples Made of Different Building Materials Designed for the Protection of Classified Information"

_materials, 2023, doi:10.3390/ma16031219_

Round 1
Reviewer 1 Report
The title of the paper is misleading. It should changed into: Drop weight testing of samples made of different building materials
- It is not relevant whether classified information has to be secured or anything else like people, installations whatever
- The introduction is much too elaborated.
- The term "overcome" is misleading. The test is towards "breakthrough"
- Line 85: So why don't you test doors or windows?
- line 87: This is not a sentence.
- line 88: Why is "pavement" a security barrier?
- Table 1: I have no idea what "wedge element of construction material" is?
- Table 1: Is thin-layer bonding mortar a reinforcement or fastener?
- Line 123: These codes are for doors, windows, curtain walls, grills and shutters rather than walls!!!
- Table 2: drop height is 5mm and 10mm. That does not make any sense.
- line 157: What are ceramic based fittings?
- 3.3 line 192: Where is the metal sheet located?
- line 272: This is no sentence.
Author Response
Dear Reviewer
Thank you for your comments and suggestions. We hope we have incorporated all your comments.
- The title of the post has been changed.
- The introduction has been changed: some sections have been removed and research in a similar area has been added
- we used the term overcome as it was a simulation of an attacker's attack, but we agreed that it was better to use the term breakthrough
- the statistics for overcoming doors and windows were for residential houses and not for rooms with classified information. We removed the statistic.
- Former lines 85 and 87 have been corrected and changed,
- wedge element of construction material is the main construction material, we redid table 1
- thin-layer bonding mortar is fastener, we redesigned table 1
- yes, EN 1629 and EN 1630 are for doors etc. but they are also used for walls, the whole testing was consulted with The National Security Authority of Slovakia.
- data in table 2 corrected
- ceramic based fittings are ceramic blocks, we have corrected this
- former line 192 : the sheet metal is located on the impact side, explanation has been added to the paper
- former line 272 changed and edited
Once again, we would like to thank you for your comments and suggestions that helped to improve the quality of our paper.
Authors of the paper.
Reviewer 2 Report
1. The main content of the paper does not involve new materials or powerful experimental methods, which leads to the lack of innovation of the paper.
2. The introduction mainly describes ‘the definition and classification of confidential information’ and ‘protective measures - walls and other building components’, while the achievements in the current research field should be summarized and analyzed.
3. From the experimental results, the phenomena are only explained, but the scientific laws behind the data are not deeply investigated.
4. The sample amount of experimental data is too few to meet the analysis.
Author Response
Dear Reviewer,
Thank you for your comments and suggestions. We believe we have completed and corrected our papper to meet all suggestions.
1-2. we have added previous achievements in the current research area, as you can see, similar tests to overcome/breach security walls were subjective as they were performed by a human and the results of these tests depend on the experience and skill of the test performer. Our tests were consulted with The National Security Authority of Slovakia.
3. for our study it is necessary to know the results of whether an attacker would penetrate a room with classified information. We have also added an explanation of the stiffness of the material and its effect on the recorded force.
4. the number of samples was consulted with the test certification authority
Once again, we would like to thank you for your comments and suggestions, on the basis of which we have corrected and completed our paper and helped us to improve the quality of our study.
Authors of the article.
Reviewer 3 Report
1. Please enhance the English language of the study
2. Please provide a background of the study by looking at past research and discussing the difference in this study compared to those.
3. the results need to be considerably revised. It's just a number of tables with minimal data.
4. Combine the discussion and conclusion and write conclusion based on each points.
Author Response
Dear reviewer,
Thank you for your comments and suggestions. We hope that we have incorporated all of your comments.
1. we have improved the English language of the study
2. we have added previous successes in the current research area, as you can see, similar tests for overcoming/breaking security walls were subjective as they were performed by a human and the results of these tests depend on the experience and skills of the test performer. Our tests were consulted with the National Security Authority of Slovak republic.
3. we have adjusted the resulting tables.
4. we combined discussion and conclusion, as well as wrote the conclusion based on the points.
Once again, we would like to thank you for your comments and suggestions, on the basis of which we have corrected and completed our paper and helped us to improve the quality of our study.
Authors of the article.
Reviewer 4 Report
· The abstract was poorly written. The executive summary of the research was not well summarized. The abstract should contain brief background/introduction, problem, objectives, methodology and findings.
· The introduction does not contain review of previous works, no research gap and research problem and research objectives
· The testing methods needs to be explained further, it is not clearly understood
· The results were not explained scientifically and not compared with other studies
· The results discussion is very shallow under section 4.0
· The conclusions need to be rewritten
· The contribution and or novelty of the studies in very low and not clear
Author Response
Dear reviewer
Thank you for your comments and suggestions. We hope that we have incorporated all of your comments.
- the abstract of the study has been revised
- similar tests and research in the field have been added to the introduction
- the testing methods have been corrected and more detail added
- the results of the study, in terms of protection of classified information, have shown that even though they are the same security class, there are large differences on the amount of force required to overcome individual walls of different construction materials
- duscussion and conclusion have been modified and supplemented
- we consider the contribution of the study to be that we have demonstrated the differences between different structural materials even when they belong to the same safety class. The results were consulted with The National Security Authority of Slovak republic.
Once again, we would like to thank you for your comments and suggestions, on the basis of which we have corrected and completed our paper and helped us to improve the quality of our study.
Authors of the article.
Round 2
Reviewer 1 Report
no further comments
Author Response
Dear Reviewer
Thank you for your review, which helped to improve the quality of our article.
Sincerely, the authors
Reviewer 2 Report
1.The shortcomings of previous studies should be added in the section of Introduction and highlights the research significance and content of this paper.
2. “The result of all the tests was that the specimen broke in two, surprisingly it broke at the brick and not at the point of the thin-layer bonding mortar, Figure 6.” Whether the reason of this phenomenon is related to the strike point?
3. Different building materials make the purpose and emphasis of structures different. The bearing capacity and intended usage should be taken into consideration for its impact resistance.
4. It is suggested to add the damage structure pictures after the second and third tests.
Author Response
Dear Reviewer
Thank you for your review, which helped to improve the quality of our article.
1- In the introduction section, we have filled in the gaps of previous studies.
2- The fact that the specimen broke in place of the brick results that the strength of the thin-layer bonding mortar is greater than the masonry strength of autoclaved aerated concrete. An explanation is added to the article.
3- All types of building materials are designed to protect Confidential level classified information. The result of the study, is that although building materials have the same purpose, they have different resistance.
4- Pictures are added.
Once again, we want to thank you for your time and review.
Sincerely, the authors
Reviewer 3 Report
The manuscript can be accepted.
Author Response

(The authors gave the same response as above.)

Reviewer 4 Report
Authors have significantly improved the manuscript. Though there are other observations
1) At the end of the introduction there is no clear statement of research problem, research gap and research objectives
2) The discussion section needs to be presented and discussed seperately from the conclusions. Please follow standard journal format
3) The result discussion is very shallow, please discuss your result findings in details.
4) There is need to check the grammar in the manuscript, avoid using phrases like "we use" throughout the manuscript. Report your work in passive form
5) The Figs in the introduction section, are they necessary ?
Author Response
Dear Reviewer
Thank you for your review, which helped to improve the quality of our article.
1- In the introduction section, we have filled in the gaps of previous studies as well as the statement of research problem, research gap and research objectives
2- The discussion and conclusion was written based on the recommendations of the other reviewers who approved it as such.
4- Grammar was checked and passive form was used.
5 - The figures will supplement the text and give a better idea of the test performance. The same have been added on the reviewer's recommendation.
Once again, we want to thank you for your time and review.
Sincerely, the authors